# Functional Characterization of Accessible Chromatin in Common Wheat

**DOI:** 10.3390/ijms25179384

**Published:** 2024-08-29

**Authors:** Dongyang Zheng, Kande Lin, Xueming Yang, Wenli Zhang, Xuejiao Cheng

**Affiliations:** 1State Key Laboratory of Crop Genetics and Germplasm Enhancement and Utilization, CIC-MCP, Nanjing Agricultural University, No.1 Weigang, Nanjing 210095, Chinakandelin2021@163.com (K.L.); 2Institute of Food Crops, Jiangsu Academy of Agricultural Sciences, Nanjing 210014, China

**Keywords:** open chromatin, DNA methylation, footprint, MHS, wheat

## Abstract

Eukaryotic gene transcription is fine-tuned by precise spatiotemporal interactions between *cis*-regulatory elements (CREs) and *trans*-acting factors. However, how CREs individually or coordinated with epigenetic marks function in regulating homoeolog bias expression is still largely unknown in wheat. In this study, through comprehensively characterizing open chromatin coupled with DNA methylation in the seedling and spikelet of common wheat, we observed that differential chromatin openness occurred between the seedling and spikelet, which plays important roles in tissue development through regulating the expression of related genes or through the transcription factor (TF)-centered regulatory network. Moreover, we found that CHH methylation may act as a key determinant affecting the differential binding of TFs, thereby resulting in differential expression of target genes. In addition, we found that sequence variations in MNase hypersensitive sites (MHSs) result in the differential expression of key genes responsible for important agronomic traits. Thus, our study provides new insights into the roles of CREs in regulating tissue or homoeolog bias expression, and controlling important agronomic traits in common wheat. It also provides potential CREs for genetic and epigenetic manipulation toward improving desirable traits for wheat molecule breeding.

## 1. Introduction

The distinct and dynamic structural organization of chromatin harbors particularly informative regulatory information, which underpins the transcription of underlying genes during normal development and environmental stress responses [1,2,3,4]. Gene transcription is spatiotemporally fine-tuned through the direct or indirect interactions between regulatory proteins and the underlying *cis*-regulatory elements (CREs). CREs in eukaryotic genomes are usually located in open chromatin regions, exhibiting more pronounced sensitivity to enzymatic cleavage/tagmentation and mechanical shearing as well relative to the immediately flanking closed chromatin [1,5,6]. A couple of high-throughput methodologies have been applied for the global mapping of chromatin accessibility (open chromatin) in yeast, humans, and plants. They include DNase-seq (DNase hypersensitive sequencing, based on DNase I) [7,8,9], ATAC-seq (assay for transposase-accessible chromatin using sequencing, based on transposase Tn5) [10,11], FAIRE-seq (formaldehyde-assisted isolation of regulatory elements coupled with sequencing) [12], and MH-seq (MNase hypersensitive sequencing) [13,14]. Compared with DNase I and Tn5, MNase has a small molecular weight with only 17 kDa [15], which allows its access to the small open space or relatively more condensed chromatin regions [14]. Thus, MH-seq can identify open chromatin regions that not only overlap with about 90% of that identified by DNase-seq and ATAC-seq, but also some specific regions that are not accessible to DNase I and Tn5 [14]. So, MH-seq has been successfully applied to identify open chromatin in plants, including maize [16], *Arabidopsis* [14], and rice [17]. Open chromatin has been generally characterized in regulating gene expression or responses to environmental changes in several plant species, including *Arabidopsis*, several grasses species, potatoes, and others [9,11,18,19,20,21,22,23,24]. However, how CREs function in the transcription of tissue-specific or subgenome-biased expression of homoeologous gene pairs in polyploids is still understudied. 

Common wheat (*Triticum aestivum* L.) is an allohexaploid containing A, B, and D subgenomes with a huge and complex genome [25]. It is formed through an interspecies hybridization between tetraploid (AABB) and *Triticum tauschii* (DD) [26,27,28]. With the availability of genome sequencing, it becomes an excellent investigation system toward understanding crop polyploidization and subgenome organization, and unveiling genetic and epigenetic mechanisms underlying the balanced or biased expression of homoeologs. A comprehensive transcriptional atlas, using a wide range of tissues, developmental stages, and cultivars, indicates that the coordinated or biased expression of wheat homoeologs may contribute to tissue development and stress responses, and the evolution of important agronomic traits, which are mediated by alterations in local chromatin organization, and subgenome neo- or sub-functionalization during or after polyploidization in common wheat [29,30,31,32]. It has been found that epigenetic mechanisms such as DNA methylation and histone modifications as well as chromatin accessibility play vital roles in controlling similar or divergent expression of homoeologs in common wheat [19,31,33], providing evidence showing *cis*- and *trans*-regulatory divergence between subgenomes involved in the biased expression of homoeologs in polyploids [34]. However, how CREs individually or in combination with epigenetic marks function in regulating homoeolog bias expression, thereby controlling important agronomic traits in common wheat is still largely unknown. 

In this study, we performed functional characterization of open chromatin in the seedling and spikelet of common wheat using MH-seq. More importantly, we provided evidence showing that interactions between DNA methylation and CREs and sequence variations in CREs play important roles in controlling the biased expression of genes, especially for genes responsible for some of the important agronomic traits in common wheat. 

## 2. Results

### 2.1. Profiling of Open Chromatin in the Seedling and Spikelet of Common Wheat Using MH-Seq 

To profile open chromatin in the seedling and spikelet, typically representing the nutritional and reproductive stage in common wheat, we generated two well-correlated biological replicates of the MH-seq data from each tissue (r = 0.95 for seedling; r = 0.93 for spikelet) (Appendix A). We merged the two biological data sets to maximize the calling of MNase hypersensitive sites (MHSs) using Model-based Analysis of ChIP-Seq (MACS2) [35]. We identified a total of 289,446 and 277,119 MHSs in the seedling and spikelet, respectively (Figure 1A). The majority of MHSs (*ca.* 63% in seedling and ca. 67% in spikelet) were common between the two tissues and the remaining ones were tissue-specific (Figure 1A, Appendix A). Representative IGV snapshots illustrate the reproducibility of MHSs between biological replicates in each tissue (Figure 1B) and subtypes of MHSs between tissues (Appendix A). As illustrated in Figure 1C, MHSs were preferentially enriched at the ends of chromosomes, and gene-rich euchromatic regions, which is similar to the previous findings in maize and wheat [19,36,37]. After plotting the normalized MH-seq read counts across ± 2 kb from the transcription start sites (TSSs) to the transcription end sites (TESs) of genes, we observed that MH-seq read density in the spikelet distributed more at the upstream of the TSSs, but less from the downstream of the TSSs to the downstream of the TESs as compared to the one in the seedling (Appendix A), reflecting differential chromatin openness between the seedling and spikelet in common wheat.

To assess the subgenomic distributions of different subtypes of MHSs, we partitioned the whole wheat genome into seven functionally annotated subgenomic domains, including promoters, exons, introns, 5′UTRs, 3′UTRs, 2 kb downstream of the TESs (downstream), and intergenic regions located at least 2 kb away from either the TSSs or the TESs of any genes. In general, the majority of MHSs in the seedling (*ca.* 85%) or spikelet (*ca.* 86%) or common MHSs (*ca.* 86%) were mainly enriched in the promoters and intergenic regions (Figure 1D). After a closer examination, we observed that the seedling MHSs distributed 2.0% more in exons and 2.9% more in intergenic regions but 4.1% less in promoters as compared to the spikelet ones. Moreover, we found that unique seedling/spikelet MHSs distributed 9.3%/1.5% more in intergenic regions, but 14.6%/4.1% less in promoters, and unique seedling MHSs exhibited 4.7% more in exons as compared to the common MHSs.

### 2.2. Biological Functions of Differential Promoter MHSs

It has been documented that tissue- or cell-type-related differential chromatin openness plays a vital role in tissue development or shaping cell identity through regulating the differential expression of related genes [1,19,38,39]. To explore the biological implications of tissue-biased open chromatin in common wheat, we first identified 48,741/22,198 seedling/spikelet-biased MHSs using MAnorm (Figure 2A). As expected, seedling- or spikelet-biased MHSs exhibited higher MHS read intensity in each subgenome (A, B, and D) than the counterpart in the spikelet or seedling (Figure 2B). We then analyzed the RNA-seq data and identified a total of 8420 upregulated and 8396 downregulated genes in the spikelet relative to the seedling. According to Gene Ontology (GO) term enrichment analyses (Appendix A), we found that upregulated genes were enriched in GO terms related to typical processes of the reproductive stage, including floral organ/flower development/morphogenesis, reproductive shoot system/tissue/meristem development, and post-embryonic morphogenesis; by contrast, downregulated genes were enriched in GO terms associated with some fundamental biological processes in the nutritional stage, including photosynthesis and some metabolic/biosynthetic processes. Thus, differentially expressed genes (DEGs) were functionally related to the typical developmental stage in each tissue. 

To explore whether tissue-related differential chromatin openness was involved in tissue-related differential gene expression, we plotted normalized MH-seq read counts across ± 2 kb from the TSSs to the TESs of DEGs in each tissue. We observed that upregulated or downregulated genes in the spikelet exhibited higher or lower MH-seq read intensity than the corresponding genes in the seedling, indicative of a direct correlation between MH-seq read intensity and the expression levels of DEGs in each tissue (Figure 2C, *p* < 2.2 × 10^−16^, KS-test). We then compared the expression levels of genes with at least 1 bp overlapping the seedling/spikelet-biased MHSs within 2 kb upstream of the TSSs. We found that genes with seedling- or spikelet-biased MHSs were more expressed than the corresponding genes in the spikelet or seedling (Figure 2D). A similar trend was observed in each subgenome (A, B, and D) (Appendix A). Furthermore, we observed that 85% of genes with spikelet-biased MHSs and 54% of genes with seedling-biased MHSs were significantly upregulated in the corresponding tissue (Appendix A). These results indicated a direct correlation between differential MH-seq read intensity and differential expression of overlapping genes in each tissue. After conducting GO term enrichment analyses, we found that genes overlapping spikelet-biased MHSs were primarily enriched in GO terms related to tissue/flower/meristem development and the regulation of developmental growth; by contrast, genes overlapping seedling-biased MHSs were mainly enriched in GO terms associated with photosynthesis, response to stimuli/chemical, and defense response (Figure 2E). These results showed that differential chromatin openness plays an important role in tissue development through regulating the expression of related genes. 

As illustrated in Appendix A, a relatively high proportion of seedling-biased MHSs overlapping transcription factors (TFs) was observed for *AP2* (14%), *bHLH* (8%), *MYB* (13%), *NAC* (9%), and *WRKY* (10%); by contrast, a relatively high proportion of spikelet-biased MHSs overlapping TFs was observed for *ABI3* (8%), *AP2* (8%), *bHLH* (9%), *MADS* (16%), *MYB* (16%), and *NAC* (8%). *ABI3* (2% vs. 8%), *AP2* (14% vs. 8%), *MADS* (4% vs. 16%), and *WRKY* (10% vs. 4%) displayed dramatic differences in the proportion of biased MHSs overlapping TFs between the seedling and spikelet (Appendix A). Interestingly, after constructing biased MHSs overlapping the TF-centered regulatory network using GENIE3 software (https://github.com/vahuynh/GENIE3, accessed on 1 August 2023) [40], we obtained a regulatory network containing *TaAG1*, a MADS-box type TF, as the hub gene (Figure 2F), indicating that function-biased MHSs in tissue development can be mediated by the TF-centered regulatory network. It has been reported that the functional defect of *ZAG1*, the homolog of *TaAG1* in maize, results in defects in the ABCDE flowering model in maize [41], indicative of the important roles of this gene in flower development. It is a widely accepted viewpoint that subgenome A, B, or D in common wheat has dominant functions in controlling grain yield, stress resistance, and grain protein content over the other two subgenomes, respectively, reflecting functional genomic asymmetry in controlling important agronomic traits in common wheat [42]. As expected, we found that the density of MHS peaks within the major dominant QTLs in subgenome A was significantly higher than the syntenic blocks in the other two subgenomes (Figure 2G, left). For instance, the higher MHS read intensity was observed in the promoter of *TracesCS5A01G473800*, a domesticated Q gene controlling important agronomic traits in wheat [43], than its homoeologs in subgenome B (*TracesCS5B01G486900*) and D (*TracesCS5D01G486600*) (Figure 2G, right).

Taken together, all the above analyses indicate that the functions of tissue-biased MHSs in regulating important agronomic traits can be mediated by overlapping the TF-centered regulatory network; thus those MHSs can serve as potential genome editing targets for bioengineering improvement of important agronomic traits in crops. 

### 2.3. Differential TF Binding between Seedling and Spikelet

The involvement of TFs in regulating gene expression is achieved through direct or indirect binding to CREs [44]. To interrogate the distinct roles of TFs in regulating differential gene expression between the seedling and spikelet or among subgenomes in each tissue, we specifically assessed footprint (FP) levels for TFs, representing the accessibility of TFs, in each tissue or each subgenome. To ensure the accuracy of the identified FPs, which were defined by the FPsore calculated by TOBIAS [45,46,47], we determined to use the top 50% of them for further analyses. As shown in Figure 3A, the TF binding sites (TFBSs) can be clearly divided into two distinct subtypes between tissues, implying distinct roles of TFs in regulating tissue development. For instance, gene *CRF4*, which mainly functions in the cold response and the regulation of N signaling in the shoot [48,49], had a higher FPscore in the seedling than in the spikelet; by contrast, gene *AP1*, which is a floral meristem identity gene [50], had a higher FPscore in the spikelet than in the seedling, reflecting the occurrence of differential binding of each TF between the two tissues (Figure 3B). After examining gene *WFZP* that has been reported to control spikelet development in common wheat [51,52], we found that its expression levels and the MNase cleavage frequency within its core motif were significantly higher in the spikelet than in the seedling (Figure 3C and Appendix A). 

To further investigate the tissue-related differential functions, we chose the top 50 TFs with the most significantly differential footprints for a dendrogram analysis. We found that the Dof TF family involved in the regulation of BR signaling, such as *COG1*, was abundant in both the seedling and spikelet. By contrast, the *AP2/ERF* TF family was preferentially abundant in the seedling, while the *MADS-box* TF family responsible for floral regulators such as *AP1*, *FLC*, *SOC1*, and *AGL27*, was significantly enriched in the spikelet (Appendix A). According to GO enrichment analyses, we found that the majority of them were enriched in GO terms associated with responses to stimuli and signals (Appendix A), and some genes were mainly involved in GO terms associated with cell growth and differentiation in the spikelet (Appendix A), and dephosphorylation in the seedling (Appendix A). Thus, these results suggest that TFs with differential footprints may be involved in regulating complex biological processes among different tissues.

We then conducted similar analyses among the triad genes. We calculated the average FPscore within the promoter regions of 1:1:1 genes. We found that the FPscore within the promoter regions of the triad genes tended to be balanced in the seedling and spikelet (Figure 3D). We obtained similar results after calculating the number of TFBSs within the promoter region of the triplet gene using TOBIAS and after downloading the number of TFBSs within the promoter region calculated by FIMO from public data [31,53] (Figure 3E, Appendix A). After associating the FPscore with the expression levels of the corresponding genes, we found that the FPscore was positively correlated with the expression levels of the corresponding genes. Moreover, we found that triad genes had higher FPscores and gene expression levels than non-triad genes (Appendix A).

Collectively, all the above analyses indicate that differential TF binding results in distinct roles of TFs between tissues or among subgenomes in common wheat.

### 2.4. Interrelationship between CHH Methylation and TFBSs 

It is well-documented that the presence of 5mC can affect the binding of TFs, thereby indirectly influencing the expression of TF target genes [54,55,56]. It inspired us to explore the possible impacts of 5mC on TFBSs in different tissues or subgenomes of wheat. We first analyzed BS-seq data generated from the spikelet and seedling, and found that global DNA methylation levels in the seedling were significantly higher than those in the spikelet (Appendix A), which matches well with the overall higher levels of chromatin openness in the spikelet than in the seedlings (Appendix A). After identifying differential DNA methylation sites (DMSs) or differential DNA methylation regions (DMRs), we found that CHH methylation exhibited a high proportion (approximately 90%) in the DMRs and DMSs, while CG and CHG only accounted for less than 10% and approximately 0.4–9%, respectively (Appendix A). In terms of each cytosine context, we found that the number of DMRs/DMSs overlapping TFBSs from the highest to the lowest order was CHH, CHG, and CG (Figure 4A). A similar trend was also observed for overlapping frequency between hyper- or hypo-DMRs/DMSs and TFBSs. Only CHH-related hyper- or hypo-DMRs exhibited significantly higher overlapping frequency with TFBSs as compared to random, and CHH-DMRs exhibited the closest distance to TFBSs (Figure 4B–D). However, the overlapping frequency of TFBSs with DMSs was lower in each cytosine context than random (Appendix A). A similar result was observed for overlapping frequency between hyper- or hypo-DMRs of each cytosine context and the ±50 bp around overlapping TFBSs (Appendix A). 

After comparing FPscores for TFs overlapping DMRs, we found that both CG- and CHG-related DMRs exhibited higher FPscore changes than CHH, consistent with previous results [56] (Figure 4E). According to GO term enrichment analyses, we observed that the closest genes around TFBSs overlapping DMRs in the seedling were mainly enriched in GO terms involved in the response to sorts of stimuli and tissue development (Appendix A), implying that DNA methylation affects the footprint of TFBSs, thereby involving them in various biological processes. Given that TFs can act as the reader of DNA methylation [57], we postulate that DMR- and DMR-TFBS-associated genes may exhibit differential gene expression during tissue development. After counting the ratio of DEGs between DMR- and DMR-TFBS-associated genes, we indeed found that the proportion of DEGs associated with DMR-TFBS was significantly higher than that of DEGs associated only with DMRs (Figure 4F,H). After calculating the number of overlaps between the biased footprints and the DMRs, we observed that CHH had the highest number of overlaps with the biased footprints, which is consistent with the previous results (Appendix A). We also found that genes associated with DMR-biased footprints always exhibited biased expression toward the biased footprint (Figure 4G, Appendix A). For example, the proportion of upregulated genes in the spikelet-biased footprints was always significantly higher than that of the downregulated genes, regardless of whether these biased footprints were overlapping the hypo- or hyper-DMRs. 

Taken together, all the above analyses indicate that CHH methylation may be the key determinant affecting the differential binding of TFs, thereby resulting in differential expression of target genes. 

## 3. Discussion

### 3.1. Involvement of CREs in Regulating Tissue-Related Differential Gene Expression and Biased Expression of Homoeologs in Common Wheat

The extent of chromatin openness is directly correlated with the expression levels of overlapping genes in multiple plant species [9,11,18,19,20,22,23,24]. Moreover, differential chromatin openness is directly correlated with differential gene expression between tissues, inducible genes, or homoeologous gene pairs [9,19,23]. Our study further provided evidence showing a direct correlation with differential chromatin openness or footprints with the differential expression of genes among subgenomes or between tissues in common wheat (Figure 2B–D, Figure 3B and Appendix A). 

Sequence variations in CREs can fine-tune the spatiotemporal or allele-specific expression of genes [58,59] in contrast to the detrimental effects of mutations in coding regions. Therefore, subtle alterations of CREs through natural mutation or artificial manipulation contribute markedly to the formation of complex traits in humans [60] and plants [61]. For instance, improvements in rice grain appearance and quality can be achieved by modifying CRE sequences in the promoter of GRAIN WIDTH 7 (*GW7*) [62]. Thus, CREs can serve as engineering hotspots for genome editing toward desirable traits or new alleles of interest for future precision crop improvement. For example, editing of the intronic enhancers of the TRIPTYCHON and CSLA10 genes results in distinct phenotypic changes in *Arabidopsis* [63]; editing of ATAC peaks in the promoter regions of *mCLE7*, *ZmFCP1*, and *ZmCLE1E5* lead to increases in kernel row number, thereby potentially improving maize yield traits [64]. Moreover, targeted *cis*-regulatory mutations unexpectedly unveil the pleiotropic effects of the *WUSCHEL* HOMEOBOX9 *(WOX9)* gene in tomato and *Arabidopsis* [65]. 

### 3.2. Impacts of CHH Methylation on Tissue-Related Differential Gene Expression in Common Wheat

It has been reported that DNA methylation exhibits two contrasting impacts on the binding of TFs in both humans and plants [57,66]. For instance, DNA methylation dramatically affects the binding of the majority of *Arabidopsis* TFs examined in vitro [56]. Rice nuclear proteins exhibit sequence-specific and CG- and/or CHG-methylation-dependent and independent binding to the promoter of rice tungro bacilliform virus [66]. Our study showed that CHH methylation exhibited higher overlapping frequency with TFBSs, and a higher percentage of biased DEGs and footprints in both the leaf and spikelet as compared to CG and CHG methylation (Figure 4B,C,E–G), indicative of cytosine context-dependent effects on the binding of putative TFs within MHSs. Therefore, our study provided evidence showing that CHH methylation does not favor the binding of certain TFs or other trans-factors, resulting in differential footprints and differential gene expression between tissues or homoeologous gene pairs in common wheat (Figure 4H). Consistently, CHH methylation may contribute to maize inbreeding depression through reducing the binding affinity of *ZmTCP* (teosinte branched1/cycloidea/proliferating cell factor), a key regulator of plant development, to its target motifs, thereby resulting in the downregulation of overlapping genes [67]. Moreover, interactions between methylated DNA motifs and TFs/proteins reveal the multifaceted roles of DNA methylation in affecting gene expression in humans and mammals [57]. Collectively, emerging evidence indicates a conserved regulatory mechanism of CHH or DNA methylation in regulating the expression of overlapping genes in eukaryotic genomes. The exact roles of CHH methylation in regulating plant gene expression depend on the binding of TFs or regulatory proteins as activators or repressors. In addition, motifs containing CHH could be epigenetically modified through the CRISPR-deactivated Cas9 (dCas9)-based methylation editing system to obtain desirable traits for crop breeding, in which dCas9 is fused with the ten-eleven translocation (TET) family of dioxygenase enzymes (TET1) or DNA methyltransferase (DNMT1), that can change the level of demethylation or methylation of the target locus [68].

## 4. Materials and Methods

### 4.1. Plant Materials

Seeds of common wheat (*Tritium aestivum* cultivar ‘Chinese Spring’-CS) and C22 were surface-sterilized via incubation in 30% H_2_O_2_ for 10 min followed by thoroughly washing with ddH_2_O five times. The seeds were pregerminated at 22 °C for 3 days. The germinated seeds were transferred into soil (1:1:3 mixture of vermiculite/perlite/peat soil) and grown in a greenhouse under 16 h light/8 h dark conditions at 22 °C. Then, 9-day-old seedlings (above-ground parts) and spikelets at the booting stage (Feeke 10) were collected for the experiments, including RNA-seq, nuclei preparation, and genomic DNA preparation. 

### 4.2. RNA-Seq Data Analyses

Public RNA-seq data (accession no. GSE139019) were re-analyzed following the published procedures [69]. Briefly, clean data were aligned to the Chinese Spring 1.0 reference genome using HISAT2 (version 2.0.5) [70]. Then, featureCounts (version 2.0.0) [71] was used to generate gene count matrices, which were used to calculate differentially expressed genes (DEGs) using DESeq2 [72] with criteria |log2| > 1 and *P-adjust* < 0.01. Only high-confidence genes were used for downstream analyses.

### 4.3. MH-Seq Library Preparation and Data Analyses

A total of 2–3 g of 9-day-old whole seedlings or spikelets from CS were collected for cross-linking in 1% of formaldehyde fixation buffer on ice (0.4 M sucrose, 10 mM Tris-HCl, pH8.0, 1 mM EDTA, 1% Formaldehyde) under vacuum for 10 min at 23–25 °C. After adding 125 mM of glycine to quench the excessive formaldehyde, the cross-linked materials were washed three times with excessive autoclave ddH_2_O. The air-dried cross-linked leaves or spikelets were ground into a fine powder in liquid nitrogen for nuclei preparation. The purified nuclei pellet was digested in MNase digestion buffer (20 mM Tris-HCl, 4 mM MgCl_2_, 1 mM CaCl_2_, 60 mM KCl, pH 7.5) with a series of MNase (M0247, 2000 gels units/μL, NEB) such as 0 units (U), 90 U, 100 U, 700 U, and 900 U, respectively. Each enzymatic reaction was stopped by adding 16 μL 0.5 M EDTA, pH 8.0. The digested nuclei were reversely cross-linked at 65 °C overnight. MNased DNA was recovered by sequentially extracting using 1 volume of phenol, phenol/chloroform mixture, and chloroform, then followed by ethanol precipitation. Then, 40–100 bp small-sized DNA fragments were recovered by running 2% agarose gel in 1xTBE buffer for MH-seq library preparation. The MH-seq library was prepared using NEBNext^®^ Ultra™ II DNA Library Prep Kit for Illumina (E7645S; New England Biolabs, Inc, Ipswich, MA, USA)and finally sequenced using the Illumina NovaSeq platform.

MH-seq raw data were processed by using fastp to remove the adapter [73]. Only read lengths greater than 50 bp were retained. All clean reads were mapped to the International Wheat Genome Sequencing Consortium (IWGSC) reference sequence (version 1.0) using bowtie2 [74] with parameter -N 1. Duplicated reads were removed using Sambamba [75]. Reads with mapping quality equal to or greater than 20 were retained for further analyses. The MACS2 program [35] (version 2.0) with parameters “--SPMR –g 14e9 -q 0.01 --nomodel” was used for calling MHS peaks. Genome-wide annotations of MHSs were performed by using ChIPseeker [76] with default parameters except for tssRegion = c (−2000, 0). MAnorm [77] was used to identify differential MHSs using the criteria: |M value| > 1 and *p* < 0.05.

### 4.4. Identification of Differential TF Binding Sites 

Sequence bias related to micrococcal nuclease digestion was corrected by using SeqOutBias [47] with the parameters as --kmer-mask = NNNNCNNNN --read-size = 150 --pdist = 50:120. The pdist = 50:120 was used to specifically process reads with insertion sizes between 50 and 120 bp. The --exact-length parameter (--read-size = 150) was not applied to strictly limit the length of the reads as 150 bp in bam. After bias correction, we performed a genome-wide scan of the JASPAR motif database [46] using TOBIAS [45] with default parameters. We chose the top 50% of the largest <spikelet>_<seedling>_change as differential footprints (https://github.com/loosolab/TOBIAS/issues/82; accessed on 1 August 2023). 

### 4.5. Bisulfite Sequencing Data Analyses 

Bisulfite sequencing (BS-seq) data were processed as previously described [33]. Methylated cytosine sites with a coverage of 3× or more reads were used for further analyses. Differentially methylated regions (DMRs)/sites (DMSs) between two samples were identified using cgmaptools [78]. The minimum 200 bp window and the maximum 500 bp window of cgmaptools were used to identify DMRs. All significant DMRs/DMSs (*p* < 0.05) were filtered by using the following criteria: the difference in methylation levels between two samples was higher than 0.4 (40%), 0.3 (30%), and 0.1 (10%) for the CG, CHG, and CHH context, respectively. Bedtools (v2.26.0) intersect [79] was used to calculate the number of overlaps between DMRs/DMSs and transcription factor binding sites (TFBSs). 

### 4.6. Resequencing Data Analyses and Linkage Disequilibrium Calculation

Vcf files of the raw resequencing data were obtained from the Wheatomics Data Center, as well as wheat vmap1.0 [80,81]. Linkage disequilibrium (LD) block regions were calculated using LDBlockShow [82]. All genes located within the candidate SNP and LD intervals were extracted followed by annotation of SNP sites within genic and promoter regions. Those genes with nonsynonymous mutations were filtered out based on the annotation results.

### 4.7. Normalization of Read Counts

The gene body and 2 kb upstream and downstream regions of the TSSs of all genes were divided into 10 bp per window for normalization. The number of mapped MH-seq reads per sliding window was first divided by the window length (10 bp), and then by the number of all mapped reads within the genome (Mb).

## 5. Conclusions

In wheat, chromatin openness coordinated with CHH methylation affecting the TF-centered regulatory network or differential binding of TFs, thereby resulting in homoeolog bias expression in seedling and spikelet development. Moreover, variations in CREs play important roles in controlling the biased expression of key genes responsible for some important agronomic traits in common wheat.

## Figures and Tables

**Figure 1 ijms-25-09384-f001:**
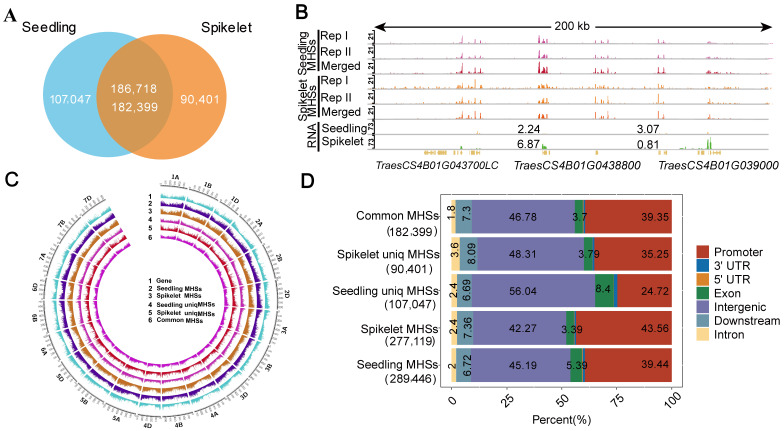
Characterization of MNase hypersensitive sites (MHSs) between the seedling and spikelet. (**A**) Venn diagram showing overlaps of MHSs between the seedling and spikelet. (**B**) A representative IGV across a 200 kb region in Chr.4B illustrating the reproducibility of MHSs between biological replicates in the seedling and spikelet. (**C**) Circle plot illustrating the distributions of subtypes of MHSs and genes in each chromosome. (**D**) Subgenomic distributions of different types of MHSs.

**Figure 2 ijms-25-09384-f002:**
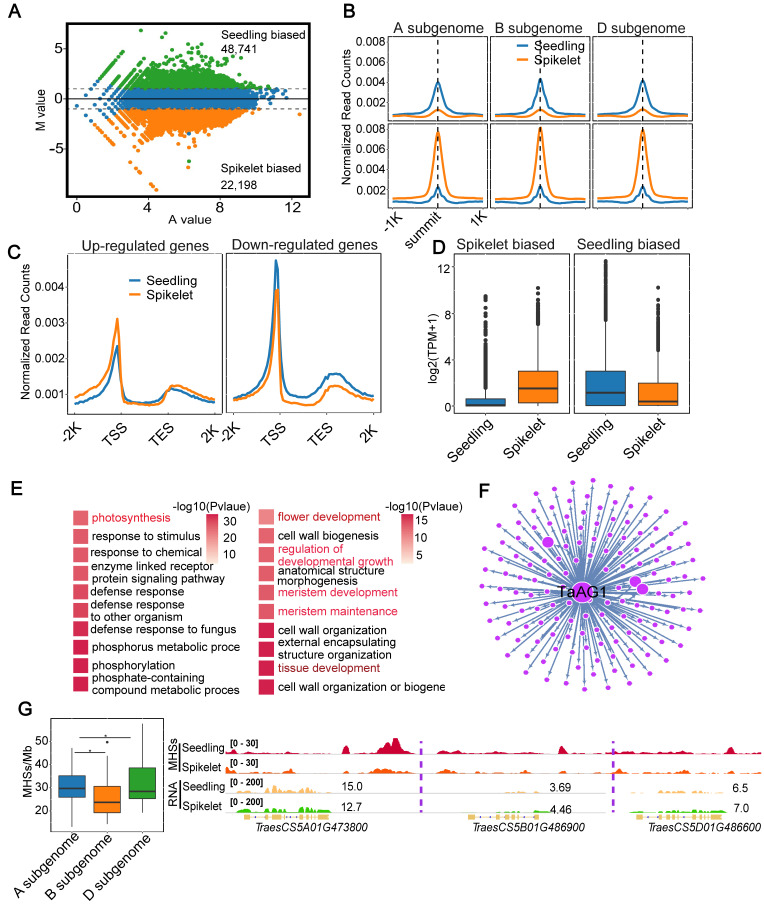
Characterization of differential MHSs and their overlapping genes between the seedling and spikelet. (**A**) MA plot showing differential MHSs in the seedling and spikelet. (**B**) Curve plot illustrating the distributions of seedling (top) and spikelet (bottom)-biased MHS read density in each subgenome. Normalized seedling (top) and spikelet (bottom)-biased MHS read counts were plotted across ±1 kb of the summit of the corresponding MHSs. (**C**) Curve plot illustrating normalized MHS read density distributed across ±2 kb from the TSSs to the TESs of upregulated and downregulated genes between the seedling and spikelet. (**D**) Comparison of expression levels of genes overlapping spikelet (left) or seedling (right)-biased MHSs. (**E**) Gene ontology (GO) enrichment analyses of genes overlapping seedling- and spikelet-biased MHSs. (**F**) GENIE3 regulatory network centered on transcription factor (TF) *TaAG1* involved in spikelet development, which was marked by spikelet-biased MHS. (**G**) The density of MHSs per Mb within major dominant QTLs controlling grain yield in subgenome A and their syntenic blocks in subgenomes B and D (left); IGV showing that the wheat domestication *Q* gene (*TracesCS5A01G473800*) had a dominant MHS and expression in subgenome A as compared to its homoeologs in subgenome B (*TracesCS5B01G486900*) and D (*TracesCS5D01G486600*) (right). Significance test was determined by Wilcoxon rank-sum test; * *p*-value < 0.05.

**Figure 3 ijms-25-09384-f003:**
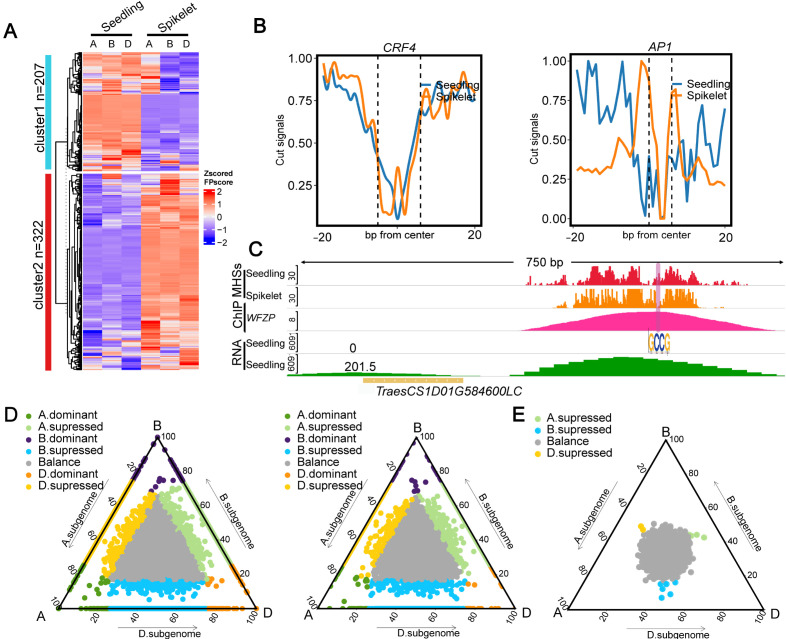
Characterization of differential footprints and their overlapping genes between the seedling and spikelet. (**A**) Heat map showing footprint (FP) scores in the spikelet and seedling; FPscores in each subgenome were plotted and clustered. (**B**) Normalized MHS read counts of seedling-biased TF CRF4 (left) and spikelet-biased TF AP1 (right). (**C**) IGV showing preferentially binding of TF WZFP in the spikelet. (**D**) TOBIAS-calculated FPscore of triad genes in the seedling and spikelet. (**E**) Number of TFBSs in the triad genes calculated by FIMO.

**Figure 4 ijms-25-09384-f004:**
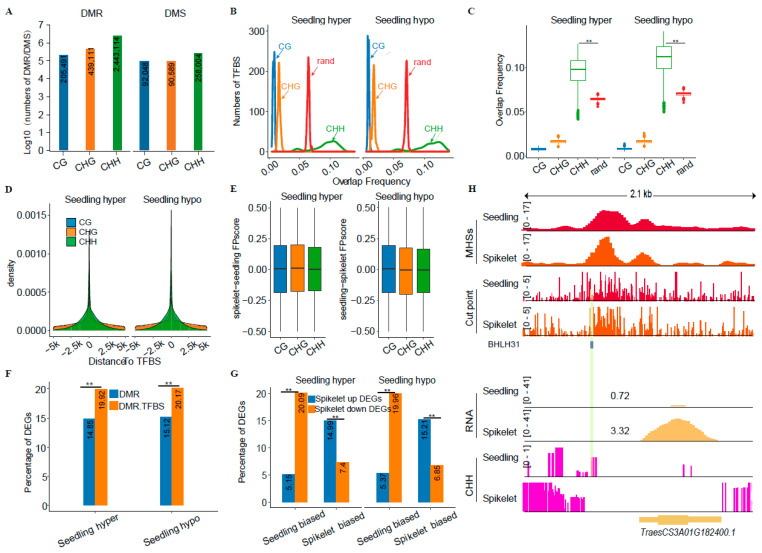
Effects of DNA methylation on TF binding. (**A**) Statistics of TF binding sites (TFBSs) overlapping DNA methylation. (**B**,**C**) Statistics of overlapping frequency between DNA methylation of each cytosine context and TFBSs; DNA methylation region overlapping frequency (**B**); Boxplot showing DNA methylation frequency (**C**); (**D**) Distance from methylated cytosines to TFBSs. (**E**) Boxplot showing the influence of DNA methylation on footprint change. (**F**) Proportion of differentially regulated genes in differential DNA methylation region (DMR)- and DMR-TFBS-associated genes. (**G**) Percentage of biased FP and DMR-related differentially expressed genes (DEGs). (**H**) An IGV across 2.1 kb in Chr. 3A illustrating the influence of the CHH methylation region on bHLH binding sites. Significance test in (**C**) was determined by Wilcoxon rank-sum test; ** *p*-value < 0.01. Significance tests in (**F**) and (**G**) were determined by Fisher’s test; ** *p*-value < 0.01.

## Data Availability

MH-seq data used in this study have been deposited in the NCBI Gene Expression Omnibus (GEO; http://www.ncbi.nlm.nih.gov/geo/; accessed on 10 August 2023) under accession number GSE186694.

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
