# Peer review of "Functional Characterization of Accessible Chromatin in Common Wheat"

_ijms, 2024, doi:10.3390/ijms25179384_

Round 1

Reviewer 1 Report

Comments and Suggestions for Authors

This study investigates how cis-regulatory elements (CREs) and DNA methylation influence gene expression in wheat. By analyzing open chromatin and DNA methylation in wheat seedlings and spikelets, the authors found that differences in chromatin openness play a significant role in tissue development and gene regulation. They also discovered that CHH methylation may impact transcription factor binding, leading to differential gene expression. Additionally, sequence variations in methylated regions affect the expression of key genes related to important agronomic traits. This research offers valuable insights into how CREs regulate gene expression and could inform strategies for wheat breeding.

Overall, the manuscript is of high quality and suitable for publication in IJMS. The authors conducted the experiments effectively and presented their results clearly. However, the manuscript may be challenging for readers unfamiliar with topics like open chromatin and DNA methylation. For instance, the authors mentioned several sequencing techniques, including MH-seq, but did not explain why MH-seq was chosen or provide a brief description of the method in the introduction.

Additionally, the manuscript contains many abbreviations related to methods, software, and other terms. Please define all abbreviations upon their first use in the manuscript. In the abstract, spell out all abbreviations, including software names, to ensure clarity for all readers. While the authors are familiar with these abbreviations, others may not be.

Lastly, the references are not formatted according to IJMS guidelines. Please check and correct the formatting.

In conclusion, I recommend the manuscript for publication after minor revisions. Congratulations on the excellent work!

Author Response

Thank you very much for taking the time to review our manuscript "Functional characterization of accessible chromatin in common wheat" (Manuscript Number: ijms-3174057). The detailed response to reviewers’ comments and the corresponding revisions in the re-submitted files.

chromatin openness play a significant role in tissue development and gene regulation. They also discovered that CHH methylation may impact transcription factor binding, leading to differential gene expression. Additionally, sequence variations in methylated regions affect the expression of key genes related to important agronomic traits. This research offers valuable insights into how CREs regulate gene expression and could inform strategies for wheat breeding.

Comment 1. Overall, the manuscript is of high quality and suitable for publication in IJMS. The authors conducted the experiments effectively and presented their results clearly. However, the manuscript may be challenging for readers unfamiliar with topics like open chromatin and DNA methylation. For instance, the authors mentioned several sequencing techniques, including MH-seq, but did not explain why MH-seq was chosen or provide a brief description of the method in the introduction.

Response: We totally agreed with the reviewer’s suggestion. Following your suggestion, we added more detailed descriptions “Compared with DNase I and Tn5, MNase has small molecular weight with only 17 kDa (Nicieza et al. 1999), which allows its access to the small open space or relatively more condensed chromatin regions (Zhao et al. 2020). Thus, MH-seq can be identify open chromatin regions that not only overlap with about 90% of that identified by DNase-seq and ATAC-seq, but also some specific regions that not accessible to DNase I and Tn5 (Zhao et al. 2020). So, MH-seq has been successfully applied to identify open chromatin in plants, including maize (Rodgers-Melnick et al. 2016), Arabidopsis (Zhao et al. 2020) and rice (Li et al. 2023)” in in page 3 line 55-63 in the revised manuscript.

Comment 2. Additionally, the manuscript contains many abbreviations related to methods, software, and other terms. Please define all abbreviations upon their first use in the manuscript. In the abstract, spell out all abbreviations, including software names, to ensure clarity for all readers. While the authors are familiar with these abbreviations, others may not be.

Response: Thank you for the suggestions. We checked all abbreviations in the manuscript, and spell out all abbreviations in the abstract and upon their first use in the manuscript

Comment 3. Lastly, the references are not formatted according to IJMS guidelines. Please check and correct the formatting.

Response: According to the suggestions, we checked and corrected the format of the references according to IJMS guidelines.

Reviewer 2 Report

Comments and Suggestions for Authors

The paper makes a valuable study using a technology that has not been massively used in wheat, therefore giving a high scientific soundness and interest to the readers.

I only have three minor suggestions for improvement.

Figure 4a: please use a logarithmic scale rather that making a gap.

Lines 356-357: Needs further explanation. What is TET1 and DNMT1? why it could be modified by CRISPR/Cas9? please develope this sentence.

Papar would gain with a conclusion indicating the major findings and the propable outcomes. 

Author Response

Thank you very much for taking the time to review our manuscript "Functional characterization of accessible chromatin in common wheat" (Manuscript Number: ijms-3174057). The detailed response to reviewers’ comments and the corresponding revisions in the re-submitted files.

Comment 1. Figure 4a: please use a logarithmic scale rather that making a gap.

Response: We thank the reviewer for pointing this out. Following your suggestions, we used a logarithmic scale to draw Figure 4a in the revised versions.

Comment 2. Lines 356-357: Needs further explanation. What is TET1 and DNMT1? why it could be modified by CRISPR/Cas9? please develope this sentence.

Response: Thank you for the suggestions. Following your suggestion, we added more detailed descriptions “the CRISPR-deactivated Cas9 (dCas9)-based methylation-editing system to obtain desirable traits for crop breeding, in which dCas9 is fused with the ten-eleven translocation (TET) family of dioxygenase enzymes (TET1) or DNA methyltransferase (DNMT1), that can change the level of demethylation or methylation of the target locus (Urbano et al. 2019)” in page 17 line 356-360 in the revised manuscript.

Comment 3. Papar would gain with a conclusion indicating the major findings and the propable outcomes.

Response: Thank you for the suggestions. Following your suggestions, we added the new section of “Conclusions” in the revised manuscript. It reads like this “In wheat, chromatin openness coordinated with CHH methylation affecting TF centered regulatory network or differential binding of TFs, thereby resulting in homoeolog bias expression in seedling and spikelet development. Moreover, variations in CREs play important roles in controlling biased expression of key genes responsible for some important agronomic traits in common wheat.” in page 17 line 359-363.

Reviewer 3 Report

Comments and Suggestions for Authors

Gene transcription is involved in precisely spatial-tempo interactions between cis-regulatory elements (CREs) and trans-acting factors. In this study, the authors characterized open chromatin coupled with DNA methylation in seedling and spikelet of common wheat. They observed differential chromatin openness occurred between seedling and spikelet, which plays important roles in tissue development through regulating expression of related genes or through TF centered regulatory network. More importantly, they provided evidence showing that interactions between DNA methylation and CREs, and sequence variations in CREs play important roles in controlling biased expression of genes, especially for genes responsible for some of important agronomic traits in common wheat.

The objectives of this study are clear, the experiments are well thought out, and the results are unequivocal. However, the figures showing the results are cumbersome, small, and difficult to understand. For example, Figure 3 has too much information, and the figure is too small to understand well. The author needs to carefully select the truly necessary figures for the entire paper and present them in an easy-to-understand manner.

Author Response

Thank you very much for taking the time to review our manuscript "Functional characterization of accessible chromatin in common wheat" (Manuscript Number: ijms-3174057). The detailed response to reviewers’ comments and the corresponding revisions in the re-submitted files.

Comment 1. The objectives of this study are clear, the experiments are well thought out, and the results are unequivocal. However, the figures showing the results are cumbersome, small, and difficult to understand. For example, Figure 3 has too much information, and the figure is too small to understand well. The author needs to carefully select the truly necessary figures for the entire paper and present them in an easy-to-understand manner.

Response: We appreciated the reviewer’s suggestions and carefully reselected the necessary information of Figures. We sent Figure 3D and Figure 4F to Supplementary File, and deleted a picture of Figure 4E, in order to present the results clearly and precisely.